# Prostate Cancer Severity in Relation to Level of Food Processing

**DOI:** 10.3390/nu15184010

**Published:** 2023-09-16

**Authors:** Salvatore Sciacca, Arturo Lo Giudice, Maria Giovanna Asmundo, Sebastiano Cimino, Ali A. Alshatwi, Giuseppe Morgia, Matteo Ferro, Giorgio Ivan Russo

**Affiliations:** 1Mediterranean Institute of Oncology (IOM), Viagrande, 95029 Catania, Italy; tsciacca42@gmail.com (S.S.); giuseppe.morgia@unict.it (G.M.); 2Department of Surgery, Urology Section, University of Catania, 95125 Catania, Italy; arturologiudice@gmail.com (A.L.G.); mariagiovannaasmundo@gmail.com (M.G.A.); ciminonello@hotmail.com (S.C.); 3Department of Food Science and Nutrition, College of Food and Agricultural Sciences, King Saud University, Riyadh 11451, Saudi Arabia; alshatwi@ksu.edu.sa; 4Department of Urology, European Institute of Oncology, IRCCS, 20141 Milan, Italy; matteo.ferro@ieo.it

**Keywords:** prostate cancer, ultra-processed foods, food processing, NOVA classification

## Abstract

Background: The level of food processing has gained interest as a potential determinant of human health. The aim of this study was to assess the relationship between the level of food processing and prostate cancer severity. Methods: A sample of 120 consecutive patients were examined for the following: their dietary habits, assessed through validated food frequency questionnaires; their dietary intake of food groups, categorized according to the NOVA classification; and their severity of prostate cancer, categorized into risk groups according to European Association of Urology (EAU) guidelines. Uni- and multivariate logistic regression analyses were performed to test the association between the variables of interest. Results: Individuals reporting a higher consumption of unprocessed/minimally processed foods were less likely to have greater prostate cancer severity than those who consumed less of them in the energy-adjusted model (odds ratio (OR) = 0.38, 95% confidence interval (CI): 1.17–0.84, *p* = 0.017 and OR = 0.33, 95% CI: 0.12–0.91, *p* = 0.032 for medium/high vs. low grade and high vs. medium/low grade prostate cancers, respectively); however, after adjusting for potential confounding factors, the association was not significant anymore. A borderline association was also found between a higher consumption of ultra-processed foods and greater prostate cancer severity in the energy-adjusted model (OR = 2.11, 95% CI: 0.998–4.44; *p* = 0.051), but again the association was not significant anymore after adjusting for the other covariates. Conclusions: The level of food processing seems not to be independently associated with prostate cancer severity, while potentially related to other factors that need further investigation.

## 1. Introduction

There is a global agreement in the scientific community that dietary factors might play an important role in population health [1]. Evidence from meta-analyses shows that diet quality is consistently linked to an increased risk of various cancers [2]. Estimates from the Global Burden of Disease study suggest that dietary risks accounted for over 10 million deaths in 2017 [3], out of which, a large share was due to cancer [4]. Moreover, dietary factors have been suggested to potentially affect cancer [5]. Recently, several researchers have suggested the hypothesis of the possible role of non-nutritional factors on human health; in fact, a growing share of commercial goods are enriched with chemical additives, including preservatives, colorants, emulsifiers, artificial sweeteners, and other various agents that have been hypothesized to affect the gut microbiota, promote oxidative stress, and act as pro-inflammatory agents, ultimately leading to an increased risk of noncommunicable diseases, possibly including cancer [6,7].

The level of processing has been considered a potential indicator of industrial and chemical additive consumption [8]. According to the NOVA classification, a food processing score [9], food products are categorized into four main groups, as follows: unprocessed, culinary processed, processed, and ultra-processed foods (UPFs) [10]. This last group includes food products that are industrially produced and heavily transformed with the addition of artificial ingredients aiming to improve their shelf life, texture, and taste [11]. These foods are typically high in calories, added sugars, unhealthy fats, salt, and other additives, and are often low in essential nutrients like fiber, vitamins, and minerals [12]. Summary evidence from nationally representative samples shows that diets with a high share of UPFs are in fact low in fiber, protein, potassium, zinc, magnesium, vitamins A, C, D, E, B12, and niacin, and rich in free sugars, total fats, and saturated fats [13]. UPFs are often marketed for their convenience, affordability, and taste, but they can have negative health effects when consumed in excess [14]. Food consumption trends show that UPF consumption ranges from an average of about 20% of daily energy intake in Mediterranean countries [15] to up to 80% in the UK [16], US [17], Canada [18], and Australia [19].

Studies have linked the regular consumption of UPFs to a range of health problems [20], including obesity, type 2 diabetes, cardiovascular disease, and mental health conditions [21,22]. Moreover, emerging evidence suggests that there may be a link between the consumption of UPFs and an increased risk of cancer. A recent study including over 100,000 participants from the *Nutrinet Santè* cohort followed for an average of five years found that a 10% increase in the proportion of UPFs in the diet was associated with a 12% increase in the risk of overall cancer [23]. Another large-scale study from the UK Biobank conducted on nearly 200,000 participants reported that UPF consumption was slightly associated with an increased risk of overall cancer [24]. Despite current data suggesting that higher UPF consumption may be a risk factor for cancer, data on specific tumor sites are rather scarce. In fact, evidence on prostate cancer specifically has been not found to be particularly convincing when examining food groups such as fruit and vegetables [25], whole grains [26], nuts and legumes [27], coffee and tea [28,29], and eggs [30] and fish [31] among animal products, with a potential detrimental association with the excess consumption of meat [32] and dairy products [33]. In contrast, studies exploring overall dietary patterns suggest that plant-based diets may play a role in preventing prostate cancer risk [2,34,35]. However, these comprehensive summaries of the literature do not take into account the level of food processing, potentially missing an important variable within a single food group or an overall dietary pattern. Thus, the aim of this study was to investigate the association between the level of food processing and prostate cancer severity.

## 2. Materials and Methods

### 2.1. Study Population

Patients were consecutively enrolled from January 2015 to December 2016 in a single institution in the municipality of Catania, southern Italy. Patients with elevated PSA and/or suspicious prostate cancer underwent a transperineal prostate biopsy (12 cores). Patients were considered eligible to be included whether they were diagnosed with clinically localized adenocarcinoma of the prostate and underwent radical retropubic prostatectomy. All the study procedures were carried out in accordance with the Declaration of Helsinki (1989) of the World Medical Association and participants provided written informed consent after accepting to participate. The study protocol was approved by the ethics committee of the referent health authority (registration number: 41/2015).

### 2.2. Clinical Data

Prostate needle biopsies were reviewed to confirm adenocarcinoma before radical prostatectomy. All biopsies received a Gleason sum [36]. Information regarding the tumor burden on the prostate biopsy was recorded as follows: greatest percentage of any single core involved by prostate carcinoma (GPC); and total overall percentage of carcinoma (TPC). TPC was calculated by adding the percentage of carcinoma on all involved cores to provide an estimate of the overall tumor burden. Tumors were staged using the TNM system, which includes extraprostatic extension and seminal vesicle invasion [37], and a second modified TNM staging system that also includes surgical margin status [38]. This later modified system classifies tumors as either being organ confined (pT2) or having adverse pathology defined as either pT3 disease (TNM system) and/or having positive surgical margins.

### 2.3. Data Collection

Demographics (including age and educational level) and lifestyle characteristics (including physical activity and smoking status) were collected. Educational level was categorized as (i) primary/secondary and (ii) tertiary (university). Physical activity level was evaluated through the International Physical Activity Questionnaires (IPAQ) [39] and based on guidelines categorized as (i) low, (ii) moderate, and (iii) high. Smoking status was categorized as (i) nonsmoker and (ii) current/ex-smoker.

### 2.4. Dietary Assessment

Dietary data were collected using two food frequency questionnaires (FFQs) validated for the population under investigation [40,41]. The long-version FFQ consisted of 110 foods and drinks referring to the participants’ diet during the last six months. Patients were specifically asked whether they had changed their diet due to course of the disease and were asked to answer the questionnaire referring to their habitual diet before the diagnosis of cancer. Participants were asked how often, on average, they had consumed foods and drinks included in the FFQ, with nine responses described as follows: ‘never’, ‘once a month’, ‘twice a month’, ‘once a week’, ‘2–3 times a week’, ‘4–5 times a week’, ‘once per day’, ‘2–3 times per day’, and ‘4–5 times per day’. Intake of food items characterized by seasonality referred to their consumption during the period in which the food was available and then was adjusted by its proportional intake in one year.

### 2.5. Ultraprocessed Food Intake

The NOVA food classification system was used to assess the intake of UPFs. The NOVA food classification system categorizes foods based on the extent and purpose of food processing [42].

The NOVA system categorizes foods into four categories:Unprocessed or minimally processed foods: these are foods that have not undergone any processing or have undergone minimal processing, such as cleaning, milling, and refrigeration (i.e., fresh fruits and vegetables, whole grains, nuts, and legumes).Processed culinary ingredients: these are substances derived from unprocessed or minimally processed foods that are used in cooking to add flavor, texture, or other culinary properties (i.e., salt, sugar, honey, vinegar, and oil).Processed foods: these are foods that have undergone more extensive processing, such as canning, freezing, drying, or fermentation, to enhance their durability and safety or to make them more convenient to use (i.e., canned fruits and vegetables, frozen vegetables, and dried fruits).Ultra-processed foods: these are foods that have undergone industrial processing to create products that are often high in sugar, salt, and unhealthy fats and are typically low in nutrients (i.e., soft drinks, candy, packaged snacks, instant noodles, and ready-to-eat meals).

### 2.6. Endpoints

Prostate cancer severity was based on its risk classification as low, intermediate, and high according to European Association of Urology (EAU) guidelines [43]. This classification is based on the grouping of patients with a similar risk of biochemical recurrence after radical prostatectomy or external beam radiotherapy. Briefly, patients were grouped into 3 groups based on the following parameters: (i) low risk, PSA <10 ng/mL, Gleason score <7, and cT1-2a; (ii) intermediate risk, PSA 10–20 ng/mL, or Gleason score = 7, or cT2b; and (iii) high risk, PSA >20 ng/mL, or Gleason score >7, or cT2c. Advanced prostate cancers were defined as any PSA, any Gleason score, and cT3-4 or cN+.

### 2.7. Statistical Analysis

The daily intake of each food group as categorized by NOVA was calculated as the proportion (%) of the total weight of foods and beverages consumed (g/d) by creating a weight ratio. This approach has been suggested to better include non-nutritional factors pertaining to food processing (i.e., additives) as compared to energy ratio. Exposure to the variables of interest was categorized based on the median cut-offs of low and high consumption of each NOVA food category group. NOVA food group intake distribution was tested for normality distribution with the Kolmogorov–Smirnov test, and it followed a slightly asymmetric normal distribution due to extreme values on the upper side. Categorical variables were presented as frequency and percentage, while continuous variables were presented as median and standard errors. Differences in frequency between groups of categorical variables were calculated using the chi-square test. The Mann–Whitney U test was used to compare differences in intakes between groups of continuous variables. The outcome was prostate cancer severity dichotomized as (i) medium/high vs. low risk and (ii) high vs. medium/low risk prostate cancers. The association between the level of intake of NOVA food groups and prostate cancer severity was calculated through logistic regression analysis providing an unadjusted model, an energy-adjusted model, and a multivariate model further adjusted also for age groups (<60 y, 60–70 y, >70 y), energy intake (kcal/d, continuous), educational status (low, high), weight status (normal, overweight, obese), smoking status (smokers, nonsmokers), and physical activity level (low, medium, high). An additional analysis was performed by entering each single variable into separate models in order to identify potential specific confounders. All reported P values were based on two-sided tests and compared to a significance level of 5%. SPSS 17 (SPSS Inc., Chicago, IL, USA) software was used for all the statistical calculations.

## 3. Results

A total of 120 prostate cancer cases were collected. Table 1 shows the characteristics of the study sample according to the intake of NOVA food processing groups. Some statistically significant differences were revealed between the groups: there was a higher proportion of never smokers among individuals reporting a higher intake of unprocessed/minimally processed foods, as well as a higher proportion of current smokers among those reporting a higher intake of processed foods (Table 1). Moreover, higher levels of education were reported among those consuming more processed culinary ingredients and a higher proportion of family history of prostatic cancer among those consuming more UPFs (Table 1).

Table 2 reports the clinical characteristics of the study participants according to their intake of NOVA food processing groups. Significant findings almost exclusively concerned the consumption of unprocessed/minimally processed foods, which was higher in individuals with less severe prostate cancer, lower GPC, and TPC (Table 2). Moreover, there was a higher proportion of patients with worse grading among those reporting higher processed food consumption (Table 2).

The mean weight ratios of NOVA food groups across categories of prostate cancer severity showed a significantly higher intake of unprocessed/minimally processed foods (including meat and poultry, fish, milk and unprocessed dairy, fruits, and legumes) in the low-grade category of patients compared to the others, as well as a slightly higher consumption of processed foods (including cheese and processed cured meats) in the high-grade category of patients (Table 3). Among UPFs, only salty snacks were consumed more by patients with higher grades of prostate cancer severity (Table 3).

Table 4 provides association measures between the consumption of foods by level of processing and the severity of prostate cancer. Individuals reporting a higher consumption of unprocessed/minimally processed foods were less likely to have greater prostate cancer severity than those reporting a lower consumption of these foods in the energy-adjusted model (OR = 0.38, 95% CI: 1.17–0.84, *p* = 0.017 and OR = 0.33, 95% CI: 0.12–0.91, *p* = 0.032 for medium/high vs. low grade and high vs. medium/low grade prostate cancers, respectively); however, after adjusting for potential confounding factors, the association was not significant anymore (Table 4). A borderline association was also found between a higher consumption of UPF and greater prostate cancer severity in the energy-adjusted model (OR = 2.11, 95% CI: 0.998–4.44; *p* = 0.051), but again the association was not significant anymore after adjusting for the other covariates (Table 4). The analysis of each single variable in separate models revealed a certain stability of results concerning the inverse relationship between unprocessed/minimally processed foods and prostate cancer severity, while its association with UPF consumption was substantially weakened after adjusting for smoking status and physical activity level (Appendix A).

## 4. Discussion

In the present study, an inverse relationship between unprocessed/minimally processed foods and prostate cancer severity was found; on the other hand, a positive association was found with UPF consumption. Both associations were no longer significant in the multivariate regression analysis adjusted for potential confounding factors. These findings suggest that the level of food processing might play a role in the severity of prostate cancer, but it may also reflect an association with other factors that are more significantly associated with disease grading.

Some studies have tried to highlight the relationship between the intake of processed foods and cancer, although with different results for various cancer sites. A recent systematic review of the literature reported that the majority of studies on the topic consist of case–control studies coinciding that an association between UPF consumption and various cancers does exist, except for prostate cancer; the outcomes confirmed in prospective studies included overall cancer risk and that of breast, colorectal, and pancreatic cancers [44]. Concerning prostate cancer, a multicentric case–control study conducted in Spain investigated the relationship between UPF consumption and the incidence of colorectal, breast, and prostate cancer. The study (which enrolled a total of 1852 colorectal cases, 1486 breast cancer cases, 953 prostate cancer cases, and 3543 healthy controls) revealed a positive association between UPF intake and colorectal cancer and breast cancer, while no association was found with prostate cancer [45]. However, a case–control study conducted in Canada on 1919 prostate cancer patients and 1991 controls aiming to assess the relationship between the level of food processing and prostate cancer reported a slight, inverse association between the consumption of unprocessed or minimally processed foods and prostate cancer; on the other hand, a higher consumption of processed foods was associated with a higher risk of overall prostate cancer [46]. Similar conclusions can be drafted from summary evidence in the literature showing that individuals adopting a more Western-style diet that may be high in processed meat, refined grains, and sugary foods are more likely to be at risk of prostate cancer [2].

The overall mechanisms underlying the present findings are potentially various but most likely depending on one another. As previously mentioned, the daily energy share of UPFs is substantially inversely related with unprocessed/minimally processed food consumption, which in turn affects overall diet quality from a nutritional point of view. There are several dietary factors that may play a role in prostate cancer risk when considering healthy or unhealthy dietary choices [47]. A variety of vitamins have been shown to play a role in apoptosis regulation (such as retinoids), increase antioxidant defenses (such as ascorbic acid), improve the immune system (such as vitamin D), and prevent DNA damage (such as folates) [48]. Moreover, a high consumption of plant-based foods would also improve the intake of polyphenols [49], which are very common secondary metabolites in the plant kingdom; these compounds are characterized by a large variety of chemical structures investigated for their potential effects on humans [50,51,52,53]. Among the many molecules investigated in common and “medicinal” plants [54], lycopene is by far the most studied compound as a preventive agent against prostate cancer [55]. This compound typically contained in tomatoes has demonstrated an ability to suppress the progression and proliferation, arrest the in-cell cycle, and induce the apoptosis of prostate cancer cells in both in vivo and in vitro studies [56]. On the other hand, a higher intake of processed foods has been associated with an increased consumption of obesogenic and proinflammatory nutrients (i.e., saturated and trans fatty acids, refined sugars, etc.) [57]. Importantly, recent reports have stressed out the possibility that UPFs could affect human health via non-nutrient pathways [58]. Processed foods may increase cancer risk via food additives and contaminants, which may lead to a rise in inflammation through various mechanisms [59]. Processed foods are chemically, biologically, and/or physically transformed, leading to the formation of processing contaminants that may have detrimental effects on human health [60]. Inflammation and oxidative stress in the tumor microenvironment have been associated with prostate cancer development and progression [61,62]. Several studies have described, through different biochemical pathways, a possible association between metabolic alterations, systemic inflammation related to metabolism, and the incidence of prostate cancer [63,64]. However, other factors involved in prostate cancer development, such as genetics and epigenetics [65,66,67,68], have been hypothesized to play a role and should be controlled for further. Overall, although the exact mechanism remains to be elucidated, there is a substantial rationale for the association between dietary factors and prostate cancer risk.

The present study has some limitations that should be considered in the interpretation of its results. The relatively small number of individuals involved in this study limits its statistical power and the generalizability of its results. Although the sample was quite homogeneous in terms of ethnicity (all patients were Caucasians) and access to healthcare (universally provided by the national health system), unmeasured residual confounding cannot be ruled out. The method used to assess dietary intake (FFQs) is subject to recall bias and participants may over- or underestimate their intake of specific food items, also depending on social desirability bias. Moreover, FFQs were not initially designed to assess the consumption of UPFs, potentially leading to their misclassification or coding mixed types of NOVA food categories within the same food item. However, all these limitations are generally common in most existing published studies and only future investigations specifically designed to test exposure to foods depending on their level of processing will be able to provide more detailed results on this matter.

## 5. Conclusions

In conclusion, scientific evidence in the literature concerning the relationship between food processing and prostate cancer is rather weak, especially when results are adjusted for confounding factors. The present study showed a potential protective role of unprocessed and minimally processed foods against prostate cancer and only a marginal role of UPFs in its development. The overall evidence suggests that improving diet quality may help lead to lower prostate cancer severity. The present findings and the results already reported in the scientific literature suggest that even though statistical significance after adjustment for various confounding factors may not be reported, future studies should look into the level of food processing when considering dietary factors associated with the risk of prostate cancer. However, further research is needed to confirm these findings and to better understand the mechanisms behind such associations.

## Figures and Tables

**Table 1 nutrients-15-04010-t001:** Demographic characteristics of the study sample according to the intake of NOVA food processing groups (*n* = 120).

	Unprocessed/Minimally Processed Foods		Processed Culinary Ingredients		Processed Foods		UPFs	
	*Low* *(n = 78)*	*High* *(n = 42)*	*Low* *(n = 74)*	*High* *(n = 46)*	*Low* *(n = 46)*	*High* *(n = 74)*	*Low* *(n = 55)*	*High* *(n = 65)*
*Age groups, n (%)*								
<60 y	6 (7.7)	3 (7.1)	6 (8.1)	3 (6.5)	4 (8.7)	5 (6.8)	5 (9.1)	4 (6.2)
60–70 y	35 (44.9)	19 (45.2)	29 (39.2)	25 (54.3)	19 (41.3)	35 (47.3)	20 (36.4)	34 (52.3)
>70 y	37 (47.4)	20 (47.6)	39 (52.7)	18 (39.1)	23 (50.0)	34 (45.9)	30 (54.5)	27 (41.5)
*Smoking status, n (%)*								
Never smokers	37 (47.4)	29 (69.0)	40 (54.1)	26 (56.5)	32 (69.6)	34 (45.9)	35 (63.6)	31 (47.7)
Current smokers	41 (52.6)	13 (31.0) *	34 (45.9)	20 (43.5)	14 (30.4)	40 (54.1) *	20 (36.4)	34 (52.3)
*Educational level, n (%)*								
Primary/secondary	61 (78.2)	33 (78.6)	62 (83.8)	32 (69.6)	30 (65.2)	64 (86.5)	47 (85.5)	47 (72.3)
Tertiary	17 (21.8)	9 (21.4)	12 (16.2)	14 (30.4)	16 (34.8)	10 (13.5) *	8 (14.5)	18 (27.7)
*Physical activity level, n (%)*								
Low	31 (40.8)	10 (23.8)	28 (37.8)	13 (29.5)	14 (31.8)	27 (36.5)	16 (29.1)	25 (39.7)
Medium	38 (50.0)	25 (59.5)	39 (52.7)	24 (54.5)	24 (54.5)	39 (52.7)	31 (56.4)	32 (50.8)
High	7 (9.2)	7 (16.7)	7 (9.5)	7 (15.9)	6 (13.6)	8 (10.8)	8 (14.5)	6 (9.5)
*BMI status, n (%)*								
Normal	24 (30.8)	17 (40.5)	24 (32.4)	17 (37.0)	17 (37.0)	24 (32.4)	19 (34.5)	22 (33.8)
Overweight	42 (53.8)	18 (42.9)	37 (50.0)	23 (50.0)	19 (41.3)	41 (55.4)	28 (50.9)	32 (49.2)
Obese	12 (15.4)	7 (16.7)	13 (17.6)	6 (13.0)	10 (21.7)	9 (12.2)	8 (14.5)	11 (19.6)
*Family history of prostatic cancer, n (%)*								
Yes	26 (33.3)	17 (40.5)	30 (40.5)	13 (28.3)	16 (34.8)	27 (36.5)	26 (47.3)	17 (26.2)
No	52 (66.7)	25 (59.5)	44 (59.5)	33 (71.7)	30 (65.2)	47 (63.5)	29 (52.7)	48 (73.8) *

* denotes *p* < 0.05. BMI categories were as follows: normal (20–25 kg/m^2^), overweight (26–29 kg/m^2^), obese (>30 kg/m^2^).

**Table 2 nutrients-15-04010-t002:** Clinical characteristics of the study sample according to the intake of NOVA food processing groups (*n* = 120).

	Unprocessed/Minimally Processed Foods		Processed Culinary Ingredients		Processed Foods		UPFs	
	*Low* *(n = 78)*	*High* *(n = 42)*	*Low* *(n = 74)*	*High* *(n = 46)*	*Low* *(n = 46)*	*High* *(n = 74)*	*Low* *(n = 55)*	*High* *(n = 65)*
*Gleason score, n (%)*								
<6	27 (34.6)	24 (57.1)	29 (39.2)	22 (47.8)	23 (50.0)	28 (37.8)	29 (52.7)	22 (33.8)
6–7	33 (42.3)	12 (28.6)	28 (37.8)	17 (37.0)	15 (32.6)	30 (40.5)	16 (29.1)	29 (44.6)
≥8	18 (23.1)	6 (14.3)	17 (23.0)	7 (15.2)	8 (17.4)	16 (21.6)	10 (18.2)	14 (21.5)
*PSA, n (%)*								
<5	30 (38.5)	27 (64.3)	33 (44.6)	24 (52.2)	28 (60.9)	29 (39.2)	32 (58.2)	25 (38.5)
5–7	26 (33.3)	12 (28.6)	22 (29.7)	16 (34.8)	14 (30.4)	24 (32.4)	15 (27.3)	23 (35.4)
>7	22 (28.2)	3 (7.1) *	19 (25.7)	6 (13.0)	4 (8.7)	21 (28.4) *	8 (14.5)	17 (26.2)
*Staging, n (%)*								
pT1	42 (53.8)	32 (76.2)	44 (59.5)	30 (65.2)	34 (73.9)	40 (54.1)	39 (70.9)	35 (53.8)
pT2	18 (23.1)	8 (19.0)	13 (17.6)	13 (28.3)	9 (19.6)	17 (23.0)	9 (16.4)	17 (26.2)
pT3	18 (23.1)	2 (4.8) *	17 (23.0)	3 (6.5) *	3 (6.5)	17 (23.0) *	7 (12.7)	13 (20.0)
*Severity, n (%)*								
Low	25 (32.1)	24 (57.1)	27 (36.5)	22 (47.8)	23 (50.0)	26 (35.1)	28 (50.9)	21 (32.3)
Intermediate	26 (33.3)	12 (28.6)	22 (29.7)	16 (34.8)	15 (32.6)	23 (31.1)	15 (27.3)	23 (35.4)
High	27 (34.6)	6 (14.3) *	25 (33.8)	8 (17.4)	8 (17.4)	25 (33.8)	12 (21.8)	21 (32.3)
*GPC, n (%)*								
<40%	23 (29.5)	26 (61.9)	26 (35.1)	23 (50.0)	24 (52.2)	25 (33.6)	29 (52.7)	20 (30.8)
40–60%	25 (32.1)	6 (14.3)	22 (29.7)	9 (19.6)	8 (17.4)	23 (31.1)	9 (16.4)	22 (33.8)
60–80%	13 (16.7)	4 (9.5)	12 (16.2)	5 (10.9)	5 (10.9)	12 (16.2)	8 (14.5)	9 (13.8)
>80%	17 (21.8)	6 (14.3) *	14 (18.9)	9 (19.6)	9 (19.6)	14 (18.9)	9 (16.4)	14 (21.5)
*TPC, n (%)*								
<40%	33 (42.3)	30 (71.4)	35 (47.3)	28 (60.9)	30 (65.2)	33 (44.6)	34 (61.8)	29 (44.6)
40–60%	29 (37.2)	10 (23.8)	29 (39.2)	10 (21.7)	12 (26.1)	27 (36.5)	12 (21.8)	27 (41.5)
>60%	16 (20.5)	2 (4.8) *	10 (13.5)	8 (17.4)	4 (8.7)	14 (18.9)	9 (16.4)	9 (13.8)
*Margins*								
No	69 (88.5)	37 (88.1)	64 (86.5)	42 (91.3)	41 (89.1)	65 (87.8)	52 (94.5)	54 (83.1)
Yes	9 (11.5)	5 (11.9)	10 (13.5)	4 (8.7)	5 (10.9)	9 (12.2)	3 (5.5)	11 (16.9)

* denotes *p* < 0.05.

**Table 3 nutrients-15-04010-t003:** Median weight ratios (WRs) and standard errors (SEs) of daily NOVA food group intake by severity of prostate cancer in the study sample (*n* = 120).

	Prostate Cancer Severity			*p*-Value
	Low	Intermediate	High	
	*WRs, median (SE)*			
Unprocessed or minimally processed foods	66.3 (1.7)	57.8 (1.9)	52.5 (1.8)	0.001
Red meat and poultry	2.7 (0.3)	3.7 (0.6)	4.7 (0.5)	0.036
Fish and seafoods	2.8 (0.3)	3.7 (0.6)	4.7 (0.5)	0.003
Milk and unprocessed dairy	4.2 (1.0)	0.9 (0.9)	9.9 (1.0)	0.042
Eggs	0.1 (0.0)	0.1 (0.0)	0.1 (0.0)	0.798
Grains and pasta	4.5 (0.5)	4.8 (0.5)	4.6 (0.7)	0.497
Fruits	22.7 (1.6)	21.6 (1.6)	17.0 (1.1)	0.003
Vegetables	10.9 (1.1)	11.0 (1.1)	6.8 (1.0)	0.070
Potatoes	0.8 (0.1)	1.4 (0.2)	1.8 (0.2)	0.003
Nuts	0.9 (0.2)	0.8 (0.2)	0.4 (0.4)	0.962
Legumes	1.6 (0.3)	1.1 (0.2)	0.6 (0.1)	<0.001
Processed culinary ingredients	1.1 (0.5)	1.2 (1.0)	0.9 (0.4)	0.423
Plant oils	0.5 (0.0)	0.4 (0.1)	0.4 (0.0)	0.186
Animal fats	0.0 (0.0)	0.0 (0.0)	0.1 (0.0)	0.174
Table sugar	0.2 (0.0)	0.3 (0.0)	0.3 (0.0)	0.469
Fruit juice (natural)	0.0 (0.5)	0.0 (1.0)	0.0 (0.4)	0.444
Processed foods	21.8 (1.5)	25.9 (1.8)	30.1 (1.8)	0.011
Breads	8.9 (0.9)	8.9 (1.1)	10.9 (0.8)	0.945
Cheese	1.7 (0.4)	2.7 (0.4)	5.2 (0.5)	0.004
Beer, wine and liquors	7.4 (1.1)	6.7 (1.3)	12.0 (1.3)	0.194
Processed meats (cured)	0.5 (0.1)	0.8 (0.2)	1.6 (0.3)	0.001
Ultraprocessed foods	7.7 (1.2)	11.1 (1.4)	11.1 (1.6)	0.113
Fast foods	0.0 (0.0)	0.0 (0.0)	0.0 (0.0)	0.733
Ultraprocessed dairy	0.6 (0.7)	0.7 (0.5)	0.9 (1.1)	0.672
Breakfast cereals	0.0 (0.1)	0.0 (0.0)	0.0 (0.1)	0.816
Biscuits, pastries, cakes	0.6 (0.2)	0.5 (0.3)	0.7 (0.1)	0.821
Confectionery and creams	0.1 (0.0)	0.1 (0.1)	0.1 (0.0)	0.072
Ice creams	1.2 (0.4)	1.0 (0.4)	1.2 (0.7)	0.628
Salty snacks	0.3 (0.1)	0.5 (0.2)	1.2 (0.2)	0.001
Carbonated soft drinks	0.0 (0.6)	0.6 (0.6)	1.8 (0.9)	0.185
Margarine	0.0 (0.0)	0.0 (0.0)	0.0 (0.0)	0.620
Distilled alcoholic drinks	0.0 (0.0)	0.0 (0.0)	0.2 (0.0)	0.142
Confectioned juices	0.0 (0.2)	0.0 (0.4)	0.0 (0.4)	0.348
Soy products	0.0 (0.5)	0.0 (0.7)	0.0 (0.0)	0.388

**Table 4 nutrients-15-04010-t004:** Odds ratios (ORs) and 95% confidence intervals (CIs) of the association between NOVA food groups by level of processing and severity of prostate cancer.

	OR (95% CI)		*p*-Value
	Low Consumption	High Consumption	
**Intermediate/high vs. low risk prostate cancers**			
*Unprocessed/minimally foods*			
Unadjusted	1	0.35 (0.16, 0.77)	0.009
Energy-adjusted	1	0.38 (1.17, 0.84)	0.017
Multivariate *	1	0.46 (0.18, 1.20)	0.111
*Processed culinary ingredients*			
Unadjusted	1	0.63 (0.30, 1.32)	0.220
Energy-adjusted	1	0.70 (0.32, 1.53)	0.371
Multivariate *	1	0.69 (0.26, 1.81)	0.444
*Processed foods*			
Unadjusted	1	1.85 (0.87, 3.91)	0.109
Energy-adjusted	1	1.69 (0.78, 3.67)	0.184
Multivariate *	1	1.39 (0.54, 3.56)	0.499
*UPFs*			
Unadjusted	1	2.17 (1.04, 4.56)	0.040
Energy-adjusted	1	2.11 (0.998, 4.44)	0.051
Multivariate *	1	1.92 (0.78, 4.75)	0.158
**High vs. intermediate/low risk prostate cancers**			
*Unprocessed/minimally foods*			
Unadjusted	1	0.32 (0.12, 0.84)	0.021
Energy-adjusted	1	0.33 (0.12, 0.91)	0.032
Multivariate *	1	0.53 (0.17, 1.59)	0.256
*Processed culinary ingredients*			
Unadjusted	1	0.41 (0.17, 1.02)	0.054
Energy-adjusted	1	0.44 (0.17, 1.12)	0.086
Multivariate *	1	0.38 (0.13, 1.18)	0.093
*Processed foods*			
Unadjusted	1	2.42 (0.98, 5.97)	0.054
Energy-adjusted	1	2.27 (0.90, 5.73)	0.082
Multivariate *	1	2.27 (0.73, 7.10)	0.159
*UPFs*			
Unadjusted	1	1.71 (0.75, 3.90)	0.202
Energy-adjusted	1	1.65 (0.72, 3.78)	0.239
Multivariate *	1	1.45 (0.56, 3.76)	0.450

* Multivariate model 1 was adjusted for energy intake (continuous, kcal/d), age groups (<60 y, 60–70 y, >70 y), BMI (normal, overweight, obese), educational level (primary/secondary, tertiary), smoking status (never, current/former), and physical activity level (low, medium, high).

## Data Availability

The data that support the findings of this study are available upon reasonable request.

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
