# Peer review of "Prostate Cancer Severity in Relation to Level of Food Processing"

_nutrients, 2023, doi:10.3390/nu15184010_

Round 1
Reviewer 1 Report
General: The introduction is lacking in any literature review of diet in the context of prostate cancer itself; plenty of literature investigating various dietary and lifestyle factors would enhance the introduction to this manuscript. The authors should narrow the focus of the introduction to prostate cancer specifically, although touching on overall cancer is still essential.
Materials: Authors are to be commended for the excellent organization of their subsections, which clearly shows how inclusion variables were assessed. However, other confounding variables such as race, insurance status, and income may be influencing PCa severity and lifestyle choices such as diet & physical activity - authors should acknowledge these confounders in discussion or attempt to account for them in their questionnaire/study design if possible.
2.7 Statistical Analysis: If the distribution is asymmetric, the authors should consider recording continuous variables as “medians” instead of “means” to enhance the findings.
Tables: Fix sizing to fit the page; the right side of the tables is cut off. Therefore, p-value results cannot be ascertained from the tables.
Discussion: Although significant results were not demonstrated in multivariable analysis, there still is inherent value in reporting specific non-significant effects in the discussion section when comparing/contrasting to existing literature.
Author Response
Reviewer 1
General: The introduction is lacking in any literature review of diet in the context of prostate cancer itself; plenty of literature investigating various dietary and lifestyle factors would enhance the introduction to this manuscript. The authors should narrow the focus of the introduction to prostate cancer specifically, although touching on overall cancer is still essential.
Author response: Thank you for the comment, we enriched the introduction provided additional information on background research relating dietary factors and prostate cancer.
Materials: Authors are to be commended for the excellent organization of their subsections, which clearly shows how inclusion variables were assessed. However, other confounding variables such as race, insurance status, and income may be influencing PCa severity and lifestyle choices such as diet & physical activity - authors should acknowledge these confounders in discussion or attempt to account for them in their questionnaire/study design if possible.
Author response: The observation raised by the reviewer is correct; however, in our specific situation we had no differences in ethnicity (all patients were Caucasian), nor insurance status (the National Health System provides health coverage for everyone going to this hospital, and since it is a single-center study there are not differences among patients). Concerning income, we used education as a proxy, because in our opinion both education and income may play a role not in the cure but rather in the prevention (and education more than income). We emphasized these concepts in the manuscript (specifically in the limitations paragraph).
2.7 Statistical Analysis: If the distribution is asymmetric, the authors should consider recording continuous variables as “medians” instead of “means” to enhance the findings.
Author response: we agree with the reviewer and provided medians and standard errors.
Tables: Fix sizing to fit the page; the right side of the tables is cut off. Therefore, p-value results cannot be ascertained from the tables.
Author response: we upload a file where all tables are clearly visible. It may be an editing issue from the journal.
Discussion: Although significant results were not demonstrated in multivariable analysis, there still is inherent value in reporting specific non-significant effects in the discussion section when comparing/contrasting to existing literature.
Author response: we added a comment in the discussion regarding the null results.
Reviewer 2 Report
1. For the dietary assessment, participants were asked how often, on average, they had consumed foods and drinks included in the FFQ, with nine responses ranging from ‘never’ to ‘4–5 times per day’. What are the nine responses? How about the results regarding the “drinks”?
2. The association between level of intake of NOVA food groups and prostate cancer severity was calculated through logistic regression analysis single-adjusted for energy intake. Why this model only adjusted with energy intake not a crude model?
3. A higher education was re-ported among those consuming more processed culinary ingredients, but the authors did not explain it in the discussion.
4. In Table 1, how about the cut-off value of BMI?
5. The mean weight ratios of NOVA food groups across categories of prostate cancer severity, please give a clearer explanation of the “mean weight ratio” (Table 3).
Minor English editing is needed.
Author Response
Reviewer 2
- For the dietary assessment, participants were asked how often, on average, they had consumed foods and drinks included in the FFQ, with nine responses ranging from ‘never’ to ‘4–5 times per day’. What are the nine responses? How about the results regarding the “drinks”?
Author response: We reported all the response options in the text as the following: ‘never’, ‘once a month’, ‘twice a month’, ‘once a week’, ‘2-3 times a week’, ‘4-5 times a week’, ‘once per day’, ‘2-3 times per day’, and ‘4–5 times per day’. The response options for the drinks are the same.
- The association between level of intake of NOVA food groups and prostate cancer severity was calculated through logistic regression analysis single-adjusted for energy intake. Why this model only adjusted with energy intake not a crude model?
Author response: We did not report a crude unadjusted morel because total energy intake is generally always considered a potential confounding factor when exploring certain food categories (such as UPFs). However, we also added an additional model in Table 4.
- A higher education was reported among those consuming more processed culinary ingredients, but the authors did not explain it in the discussion.
Author response: Thanks for the note, we added a comment on that in the discussion.
- In Table 1, how about the cut-off value of BMI?
Author response: We added the cut-offs values of BMI.
- The mean weight ratios of NOVA food groups across categories of prostate cancer severity, please give a clearer explanation of the “mean weight ratio” (Table 3).
Author response: Thanks for the note, there was a mistake in the manuscript that we amended and we better defined mean weight ratio in the method section.
Comments on the Quality of English Language
Authors should also review the abbreviations; for example, UPF appears in the abstract without having appeared in full.
Author response: Thanks for the note, we double checked the manuscript for missing explanation of abbreviations.
Reviewer 3 Report
The “Prostate Cancer Severity in Relation to Level of Food Processing” manuscript introduces a current and pertinent theme in literature.
The association between the level of food processing and cancer is a topic that has been studied over the last few years. The authors found some evidence of this association in this article, but the model was not explanatory due to some confounding factors. I had some doubts when I read the article:
- What were the confounding variables in the model?
- Did the authors work on models with compound variables? For example, UPF x Obesity or UPF x smokers? In other words, it would be interesting to run composite variables to understand whether these variables are more explanatory for the model.
Authors should also review the abbreviations; for example, UPF appears in the abstract without having appeared in full
Author Response
The “Prostate Cancer Severity in Relation to Level of Food Processing” manuscript introduces a current and pertinent theme in literature.
The association between the level of food processing and cancer is a topic that has been studied over the last few years. The authors found some evidence of this association in this article, but the model was not explanatory due to some confounding factors. I had some doubts when I read the article:
- What were the confounding variables in the model?
Author response: The confounding variables in the model are listed in the statistical analysis section.
- Did the authors work on models with compound variables? For example, UPF x Obesity or UPF x smokers? In other words, it would be interesting to run composite variables to understand whether these variables are more explanatory for the model.
Author response: Thanks for the note, in order to not overcrowd the actual table in the manuscript, we provided an additional table with the analyses required by the reviewer, added the description in the statistical analysis section, and described them in the results/discussion sections.
Round 2
Reviewer 2 Report
The authors have addressed most of the issues raised by reviewers.
The quality of the English Language is acceptable.
Reviewer 3 Report
The authors answered the last questions. I recommend accepting the manuscript for publication.